# Pitfalls of Epistemic Uncertainty Quantification through Loss Minimisation

**Viktor Bengs**[a]**, Eyke Hüllermeier**[a,b]
[a]Institute of Informatics, University of Munich (LMU)
[b]Munich Center for Machine Learning
viktor.bengs@lmu.de, eyke@lmu.de

**Willem Waegeman**
Department of Data Analysis and Mathematical Modeling
Ghent University
Willem.Waegeman@UGent.be

## Abstract

Uncertainty quantification has received increasing attention in machine learning in the recent past. In particular, a distinction between aleatoric and epistemic uncertainty has been found useful in this regard. The latter refers to the learner's (lack of) knowledge and appears to be especially difficult to measure and quantify. In this paper, we analyse a recent proposal based on the idea of a second-order learner, which yields predictions in the form of distributions over probability distributions. While standard (first-order) learners can be trained to predict accurate probabilities, namely by minimising suitable loss functions on sample data, we show that loss minimisation does not work for second-order predictors: The loss functions proposed for inducing such predictors do not incentivise the learner to represent its epistemic uncertainty in a faithful way.

## 1   Introduction

The notion of uncertainty has received increasing attention in machine learning (ML) research in the last couple of years, especially due to the steadily increasing relevance of ML for practical applications. In fact, a trustworthy representation of uncertainty should be considered as a key feature of any ML method, all the more in safety-critical domains such as medicine [27, 17] or socio-technical systems [25, 26].

In the literature, two inherently different sources of uncertainty are commonly distinguished, referred to as *aleatoric* and *epistemic* [9]. While the former refers to variability due to inherently random effects, the latter is uncertainty caused by a lack of knowledge and hence relates to the epistemic state of an agent. Thus, epistemic uncertainty can in principle be reduced on the basis of additional information, while aleatoric uncertainty is non-reducible.

The distinction between different types of uncertainty and their quantification has also been adopted in the recent ML literature [21, 13], and various methods for quantifying aleatoric and epistemic uncertainty have been proposed [10]. In the context of supervised learning, the focus is typically on *predictive uncertainty*, i.e., the learner's uncertainty in the outcome $y \in \mathcal{Y}$ given a query instance $\boldsymbol{x} \in \mathcal{X}$ for which a prediction is sought. The aleatoric part of this uncertainty is due to the supposedly stochastic nature of the dependence between instances and outcomes, e.g. due to wrong class annotations or a lack of informative features. Therefore, the "ground-truth" is a conditional probability distribution $p(\cdot \,|\, \boldsymbol{x})$ on $\mathcal{Y}$, i.e., each outcome $y$ has a certain probability $p(y \,|\, \boldsymbol{x})$ to occur. Even with full knowledge about $p(\cdot \,|\, \boldsymbol{x})$, the outcome cannot be predicted with certainty.

36th Conference on Neural Information Processing Systems (NeurIPS 2022).

Obviously, the learner does not have full knowledge of $p(\cdot \,|\, \boldsymbol{x})$. Instead, it produces a "guess" $\hat{p}(\cdot \,|\, \boldsymbol{x})$ on the basis of the sample data provided for training. Broadly speaking, epistemic uncertainty is uncertainty about the true probability and hence the discrepancy between $p$ and $\hat{p}$. This (second-order) uncertainty can be captured and represented in different ways. In the Bayesian approach, for example, the learner's uncertainty is represented by the posterior predictive distribution, which results from the posterior on the hypothesis space [7, 6]; in other words, uncertainty about the probabilistic predictor is translated into uncertainty about the prediction in a point $\boldsymbol{x}$. Alternatively, the authors of [12] propose to capture the learner's epistemic uncertainty by means of a kind of meta-learner, which seeks to predict the difference between the total uncertainty (expected loss of the actual predictor) and the aleatoric uncertainty (expected loss of the Bayes predictor); this excess loss is then equated with the learner's epistemic uncertainty.

Yet another quite popular idea is to estimate uncertainty in a more direct way, and to let the learner itself predict, not only the target variable, but also its own uncertainty about the prediction [22, 18, 19, 20, 3, 11, 15]. For example, instead of predicting a probability distribution $\hat{p}(\cdot \,|\, \boldsymbol{x})$, the learner may predict a second-order distribution in the form of a distribution of distributions or a set of distributions [23]. The learner's epistemic uncertainty is then represented by the "peakedness" of the former and the size of the latter.

Either way, looking at existing methods, it appears that epistemic uncertainty is difficult to quantify in an objective manner. In fact, most methods dispose of parameters or other means that directly influence the amount of uncertainty, rendering epistemic uncertainty quantification arbitrary to a large extent. At second glance, this is perhaps not very surprising, because, unlike aleatoric uncertainty, epistemic uncertainty is not a property of the data or the data-generating process, and there is nothing like a "ground truth" epistemic uncertainty. Instead, the learner's uncertainty is influenced in various ways, for example by underlying model assumptions and the prior knowledge it is equipped with. For example, enlarging the learner's hypothesis space and allowing it to fit the data in a more flexible way will increase its epistemic uncertainty [10]. This is comparable to Bayesian inference, where the informedness of the posterior strongly depends on the informedness of the prior (unless the sample size is very large).

In this paper, we demonstrate the difficulty of epistemic uncertainty quantification for one of the approaches that have recently been proposed in the literature, namely, the direct prediction of second-order distributions through empirical loss minimisation. Analysing this approach in a critical way, we isolate problems questioning its practicability (Section 3). These concerns are substantiated by formal results showing that the approach does not behave as it is supposed to do (Section 4). These formal results are supported by simulations on synthetic data sets (Section 5)

## 2 Setting and Notation

Considering classification as a learning task, we assume a standard setting with instance space $\mathcal{X}$, label space $\mathcal{Y} = \{y_1, \dots, y_K\}$, and training data $\mathcal{D} = \big\{\big(\boldsymbol{x}^{(n)}, y^{(n)}\big)\big\}_{n=1}^{N} \subset \mathcal{X} \times \mathcal{Y}$. As usual, we also assume that the data is generated i.i.d. according to an underlying joint probability measure $P$ on $\mathcal{X} \times \mathcal{Y}$, i.e., each $z^{(n)} = (\boldsymbol{x}^{(n)}, y^{(n)})$ is a realisation of $Z = (X, Y) \sim P$. Correspondingly, each instance $\boldsymbol{x} \in \mathcal{X}$ is associated with a conditional distribution $p(\cdot \,|\, \boldsymbol{x})$ on $\mathcal{Y}$, such that $p(y \,|\, \boldsymbol{x})$ is the probability to observe label $y$ as an outcome given $\boldsymbol{x}$.

Let $\mathbb{P}(\mathcal{Y})$ denote the set of probability distributions on $\mathcal{Y}$, which can be identified with the $K$-simplex

$$\Delta_K := \big\{\boldsymbol{\theta} = (\theta_1, \dots, \theta_K) \in [0, 1]^K \,|\, \|\boldsymbol{\theta}\|_1 = 1\big\} \tag{1}$$

of probability vectors $\boldsymbol{\theta}$, each of which identifies a categorical distribution $\mathrm{Cat}(\boldsymbol{\theta})$. Slightly abusing notation, we shall not distinguish between vectors and distributions (functions); for example, we write $\boldsymbol{\theta}(y)$ instead of $p(y)$, which means that $\boldsymbol{\theta}(y) = \theta_k$ if $y = y_k$. A summary of the notation used in this paper is given in Section A.

### 2.1 Learning Predictive Models

Suppose a *hypothesis space* $\mathcal{H}$ to be given, where a hypothesis $h \in \mathcal{H}$ is a mapping $\mathcal{X} \longrightarrow \Delta_K$. Thus, a hypothesis maps instances $\boldsymbol{x} \in \mathcal{X}$ to probability distributions on outcomes. In standard supervised learning, the goal of the learner is, based on a loss function $L : \Delta_K \times \mathcal{Y} \longrightarrow \mathbb{R}$, to

induce a hypothesis (predictive model) with low risk (expected loss)

$$R(h) := \int_{\mathcal{X} \times \mathcal{Y}} L(h(\boldsymbol{x}), y) \, d \, P(\boldsymbol{x}, y) \ . \tag{2}$$

The choice of a hypothesis is commonly guided by the empirical risk

$$R_{emp}(h) := N^{-1} \sum\nolimits_{n=1}^{N} L\big(h(\boldsymbol{x}^{(n)}), y^{(n)}\big) \ , \tag{3}$$

i.e., the performance of a hypothesis on the training data. However, since $R_{emp}(h)$ is only an estimation of the true risk $R(h)$, the empirical risk minimiser $\hat{h} := \mathrm{argmin}_{h \in \mathcal{H}} R_{emp}(h)$ (or any other predictor) favored by the learner will normally not coincide with the true risk minimizer (Bayes predictor) $h^* := \mathrm{argmin}_{h \in \mathcal{H}} R(h)$. Correspondingly, there remains (epistemic) uncertainty regarding $h^*$ as well as the approximation quality of $\hat{h}$ (in the sense of its proximity to $h^*$) and the predictions $\hat{p}(\cdot \,|\, \boldsymbol{x}) = \hat{h}(\boldsymbol{x})$ produced by this hypothesis.

## 2.2 Learning Level-2 Predictors

Here, motivated by recent methods for uncertainty quantification, we are interested in learning a *second-order* or *level-2* predictor that is able to properly represent its own (epistemic) uncertainty, that is, a hypothesis of the form

$$H : \mathcal{X} \longrightarrow \Delta_K^{(2)} , \tag{4}$$

where $\Delta_K^{(2)} = \mathbb{P}(\mathbb{P}(\mathcal{Y}))$ denotes the set of second-order distributions, i.e., probability distributions on $\Delta_K$. If $H(\boldsymbol{x}) = Q \in \Delta_K^{(2)}$, then $Q$ assigns a probability (density) $Q(\boldsymbol{\theta})$ to each distribution $\boldsymbol{\theta} \in \Delta_K$, and the more certain the learner about the true distribution, the more concentrated $Q$ is.

If second-order or level-2 distributions are Dirichlet (cf. Appendix B), then every $Q \in \Delta_K^{(2)}$ is identified by a parameter vector $\boldsymbol{\alpha} \in \mathbb{R}_+^K$, and hence $\Delta_K^{(2)}$ with the parameter space $\mathbb{R}_+^K$. Thus, hypotheses are of the form $H : \mathcal{X} \longrightarrow \mathbb{R}_+^K$, where $H(\boldsymbol{x}) = \boldsymbol{\alpha}(\boldsymbol{x}) = (\alpha_1(\boldsymbol{x}), \dots, \alpha_K(\boldsymbol{x}))$. Thus, the Dirichlet parameters $\alpha_k$ are expressed as a function of instances, and this dependence is supposedly captured by the underlying hypothesis space $\mathcal{H}$.

## 2.3 Special Scenarios

For ease of exposition, we shall specifically look at the following scenarios, which are important special cases of the general setting:

- Binary classification: This is the case $K = 2$, where $\mathcal{Y} = \{0, 1\}$ consists of only two classes. The conditional distribution $p(\cdot \,|\, \boldsymbol{x})$ is now determined by the probability vector $\boldsymbol{\theta} = (\theta_0, \theta_1)$, i.e., by the two probabilities $\theta_0 = p(0 \,|\, \boldsymbol{x})$ that $Y = 0$ and $\theta_1 = p(1 \,|\, \boldsymbol{x})$ that $Y = 1$. As these sum up to 1, the learning problem effectively comes down to inference about the ground truth probability $\theta_1(\boldsymbol{x}) = p(1 \,|\, \boldsymbol{x})$ of the positive class, and hence about the parameter of a Bernoulli distribution.
- Coin tossing: This is a further simplification of the binary case, which can be seen as learning without instance space. Or, equivalently, we may assume an instance space $\mathcal{X} = \{\boldsymbol{x}_0\}$ consisting of only a single instance, which is observed over and over again (and can therefore be ignored, as it does not carry any information). Like in the binary case, learning comes down to estimating a ground truth Bernoulli distribution with parameter $\theta_1 \equiv p(1 \,|\, \boldsymbol{x})$, with the difference that this parameter no longer depends on $\boldsymbol{x}$.

As an important difference, note that the coin tossing scenario provides *several* observations pertaining to a single parameter $\theta_1$, i.e., several realisations of the same Bernoulli random variable, and hence information in the form of *relative frequencies*. This yields a solid statistical basis for estimating $\theta_1$. In the more general (machine learning) scenario, one can assume that at most a single observation is made in a point $\boldsymbol{x} \in \mathcal{X}$, which, in principle, does not allow for estimating a probability [1]. The common way out is to make *regularity assumptions*, so that outcomes observed for nearby points are also deemed representative for $\boldsymbol{x}$ to some extent. This becomes especially explicit for local learning methods such as decision trees and nearest neighbours, where the class probabilities are assumed to be constant within a certain region of the instance space — effectively, learning in such a region is thus again reduced to the coin tossing scenario.

| Level | Representation | Loss | Ground truth |
|---|---|---|---|
| Level 2 (epistemic) | $Q \in \Delta_K^{(2)}, \boldsymbol{\alpha} \in \mathbb{R}_+^K$ | $L_2(Q, y)$ | — |
| Level 1 (aleatoric) | $p \in \Delta_K, \boldsymbol{\theta} \in [0,1]^K$ | $L_1(\boldsymbol{\theta}, y)$ | $\boldsymbol{\theta}^*$ |
| Level 0 (observational) | $y_k \in \mathcal{Y}$ | $L_0(\hat{y}, y)$ | $y^*$ |

## 3 Learning Level-2 Predictors

In supervised learning, (level-0) samples drawn from the categorical random variable $Y \sim \mathrm{Cat}(\boldsymbol{\theta})$ are made available (explicitly or implicitly) as a basis for learning the level-1 distribution $\boldsymbol{\theta}$. However, corresponding samples are actually not provided for the level-2 distribution $Q$. In principle, to estimate $Q$, for example a Dirichlet parameter $\boldsymbol{\alpha}$, observations of realisations of that distribution would be needed, i.e., a sample in the form of probability vectors $\boldsymbol{\theta}^{(1)}, \ldots, \boldsymbol{\theta}^{(N)} \sim \mathrm{Dir}(\boldsymbol{\alpha})$. Given data of that kind, $\boldsymbol{\alpha}$ could be estimated by means of maximum likelihood maximisation or any other statistical method. However, such data cannot exist even in principle, because $\boldsymbol{\theta}$ (resp. $\boldsymbol{\theta}(\boldsymbol{x})$) is supposedly constant. This suggests that (probabilistic) learning on the epistemic level cannot be frequentist in nature, unlike learning (about $\boldsymbol{\theta}$) on the aleatoric level. Instead, it appears that learning on the epistemic level is necessarily Bayesian and requires a prior, which then of course has an influence on the degree of (epistemic) uncertainty. We shall return to this point in Section 3.2.

In light of this, one may also wonder whether it is possible to learn a level-2 predictor (4) in the "classical" way through loss minimisation, just like a level-1 predictor. In other words, is it possible to specify a level-2 loss function

$$L_2 : \Delta_K^{(2)} \times \mathcal{Y} \longrightarrow \mathbb{R}_+ \tag{5}$$

comparing level-2 predictions $Q(\boldsymbol{x})$ with level-0 observations $y$, so that minimising $L_2$ on the training data $\mathcal{D}$ yields a "good" level-2 predictor? This is the basic idea of *direct* epistemic uncertainty prediction [22, 18, 19, 20, 3, 11, 15].

Before the above question can be addressed, we need to clarify what we mean by "good" predictor. For level-1 predictors, this question is answered through the notion of *proper scoring rules* [8]. These are loss functions $L_1$ that compare (first-order) probability distributions $\boldsymbol{\theta}$ with outcomes $Y$ and guarantee that the loss minimiser coincides with the ground truth distribution $\boldsymbol{\theta}^*$ (at least asymptotically). In other words, the expected loss

$$\mathbb{E}_{Y \sim \boldsymbol{\theta}^*} L_1(\boldsymbol{\theta}, Y) \tag{6}$$

is minimised by predicting $\hat{\boldsymbol{\theta}} = \boldsymbol{\theta}^*$. Consequently, proper scoring rules provide a loss-minimising learner with an incentive to predict the true distribution, e.g. log-loss or quadratic loss [16].

This concept cannot be transferred directly to the case of level-2 predictions, simply because, as already mentioned, there is no ground-truth $Q^*$. Instead, a level-2 representation is a representation of the learner's *belief* about the level-1 ground truth $\boldsymbol{\theta}^*$. So what exactly should be the purpose of a loss $L_2$? In this regard, one should first of all notice that a (level-1) loss function may in general serve different purposes:

- A loss can be a *target loss*, which essentially means that it is determined by the application: $L(\hat{\boldsymbol{\theta}}, y)$ or $L(\hat{y}, y)$ is the *real* cost caused by the level-1 prediction $\hat{\boldsymbol{\theta}}$ or the level-0 prediction $\hat{y}$ when the ground-truth is $y$.
- A loss can be a *surrogate loss*, which means that it serves an auxiliary purpose and is used in a more indirect way: minimising the loss helps to achieve the actual goal, such as probability estimation in the case of proper scoring rules. Another example is the use of the hinge loss as a surrogate in classification; being convex and continuous, it simplifies training, although the true target is the 0/1 loss.

A level-2 loss should arguably be more of the second kind, and its purpose should be twofold: It should incentivise the learner to make predictions that are *correct* in the sense of assigning high probability to the ground-truth $\boldsymbol{\theta}^*$, and at the same time *faithful* in the sense of appropriately expressing the learner's (epistemic) uncertainty. The second point appears to be specifically delicate, due to the lack of an objective ground-truth. In fact, one may wonder how it should be possible to evaluate the faithfulness of a prediction on the basis of empirical data in the form of observed class labels. Besides, there is of course a risk of *imposing* the epistemic uncertainty on the learner, i.e., of incentivising a representation of uncertainty only because it appears favourable from a loss minimisation perspective.

## 3.1 Averaging Level-1 Losses

Several authors have proposed the minimisation of an empirical loss of the form

$$L = N^{-1} \sum_{n=1}^{N} L_2\big(Q^{(n)}, y^{(n)}\big) , \tag{7}$$

$$L_2(Q, y) = \mathbb{E}_{\boldsymbol{\theta} \sim Q} L_1(\boldsymbol{\theta}, y) , \tag{8}$$

where $Q^{(n)} = H(\boldsymbol{x}^{(n)})$. Thus, an individual prediction $Q$ is penalised in terms of the *expected* level-1 loss, with the expectation taken over the realisations of $\boldsymbol{\theta}$. For example, in [3] the level-1 loss is defined in terms of the cross entropy $\mathrm{CE}(\boldsymbol{\theta}, y^{(n)})$ and (8) is called the *uncertain cross entropy loss*, while in the evidential networks approach [22], $L_1$ is the quadratic loss (Brier score).

This approach suggests an interpretation in terms of a "matching learner" which samples predictions $\boldsymbol{\theta} \sim Q$ at random according to its current belief $Q$. But why should a learner predict in this way? For a learner seeking to minimise expected loss, wouldn't it be better to predict the most likely probability $\boldsymbol{\theta}$ throughout? Indeed, if $L_1$ is a convex loss, then Jensen's inequality implies that

$$L_1(\bar{\boldsymbol{\theta}}, y) \leq \mathbb{E}_{\boldsymbol{\theta} \sim Q} L_1(\boldsymbol{\theta}, y) , \tag{9}$$

where $\bar{\boldsymbol{\theta}} = \mathbb{E}_{\boldsymbol{\theta} \sim Q} \boldsymbol{\theta}$ is the expected level-1 prediction. As can be seen, for the learner it is better to predict the expected probability rather than sampling a probability $\boldsymbol{\theta}$ at random. As a consequence, the learner will have a tendency to peak the level-2 distribution, thereby pretending full certainty rather than representing uncertainty in an honest way. The same happens in the case of a concave loss[1], although here the tendency is toward extreme predictions.

For illustration, consider the coin tossing scenario (i.e., estimation of a constant Bernoulli parameter $\theta_1$), and suppose that $N_1$ positive and $N_0$ negative examples have been observed, hence $N = N_0 + N_1$ samples in total. Then, assuming level-2 predictions $Q$ in the form of Dirichlet distributions $\mathrm{Dir}(\boldsymbol{\alpha})$, the loss minimiser of (7) is given by the Dirichlet peaked at $\theta_1 = N_1/N$, i.e., $\boldsymbol{\alpha} = (c\,(1 - \theta), c\,\theta)$ for $c \to \infty$. So strictly speaking, the loss minimiser is not even well defined. But perhaps more important than this technical issue is the observation that the learner will always pretend full certainty about $\theta_1$, regardless of the sample size.

The case of a concave loss $L_1$ leads to even more questionable predictions. Here, one obtains the Dirichlet peaked at $\theta_1 = 0$ resp. $\theta_1 = 1$, i.e., $\boldsymbol{\alpha} = (c, 0)$ resp. $\boldsymbol{\alpha} = (0, c)$ for $c \to \infty$, in the case where $N_0 > N_1$ resp. $N_0 < N_1$. Thus, the learner may even pretend full certainty about a distribution (e.g., $\theta_1 = 1$) although that distribution is definitely excluded as the ground truth (e.g., because $N_0 > 0$).

Note that a very similar effect can be observed "one level below" (level-1 loss as expected level-0 loss): Consider a level-1 learner holding a probability $p \in \Delta_K$. This learner could be assessed by averaging over level-0 losses, i.e.,

$$L_1(p, y) = \sum_{k=1}^{K} p(y_k) L_0(y_k, y) , \tag{10}$$

where $L_0$ could be the 0/1 loss. Again, even if $p$ is a proper expression of the learner's aleatoric uncertainty, it will not be the minimiser of (10) and hence not be delivered by a loss-minimising learner. Instead, it will be best to predict the mode $y^*$ of $p$.

## 3.2 Bayesian Losses and Regularisation

Adopting a Bayesian perspective, a level-2 prediction $Q$ would naturally be seen as the posterior uncertainty about $\boldsymbol{\theta}$ given the data, i.e.,

$$Q(\boldsymbol{\theta}) = \mathrm{P}(\boldsymbol{\theta} \,|\, y) \propto \mathrm{P}(y \,|\, \boldsymbol{\theta}) \cdot \mathrm{P}(\boldsymbol{\theta}) = \boldsymbol{\theta}(y) \cdot Q_0(\boldsymbol{\theta}) ,$$

where $Q_0$ is a prior on $\Delta_K$. Roughly speaking, (8) only captures the first part on the right-hand side, the likelihood, but not the second part, the prior. Once again, this explains why putting all mass on a single $\boldsymbol{\theta}$, namely the one with the maximum likelihood, is a plausible strategy. This will of course be avoided by proper Bayesian inference, because the posterior will then be a compromise between the likelihood and the prior $Q_0$, which serves as a regulariser.

---

[1]Albeit not very common in machine learning, such losses can be useful for reasons of robustness [5].

Interestingly, a close connection between Bayesian inference and learning through loss minimisation has been established in [2]. There, it is shown that a posterior $Q$ is the minimiser of the loss

$$L(Q, y \mid Q_0) = \int L_1(\boldsymbol{\theta}, y) \, Q(d\boldsymbol{\theta}) + d_{KL}(Q, Q_0) = \mathbb{E}_{\boldsymbol{\theta} \sim Q} \, L_1(\boldsymbol{\theta}, y) + d_{KL}(Q, Q_0) \,, \qquad (11)$$

where $d_{KL}$ denotes the KL-divergence and $L_1(\boldsymbol{\theta}, y)$ is the loss caused by $\boldsymbol{\theta}$ on the observation $y$. The latter is given by the logarithmic (or self-information) loss $-\log f(y \mid \boldsymbol{\theta})$ when the data is known to be generated by the distribution with density $f(y \mid \boldsymbol{\theta})$, in which case (11) coincides with standard Bayesian inference. However, $L_1$ can also be another loss in case the data-generating process is not known. The authors consider (11) as a Bayesian version of conventional empirical risk minimisation, when the interest is on probability measures $Q$ on $\Delta_K$ rather than point estimates $\boldsymbol{\theta} \in \Delta_K$. The general solution can be shown to be of the following form:[2]

$$Q^*(\boldsymbol{\theta}) = \underset{Q}{\operatorname{argmin}} \, L(Q, y \mid Q_0) = \frac{\exp(-L_1(\boldsymbol{\theta}, y)) Q_0(\boldsymbol{\theta})}{\int \exp(-L_1(\boldsymbol{\theta}, y)) Q_0(d\boldsymbol{\theta})} \,. \qquad (12)$$

Using the loss (11) in (7) yields the level-2 empirical loss

$$L = N^{-1} \sum_{n=1}^{N} L_2\big(Q^{(n)}, y^{(n)}\big) \,, \quad \text{where} \qquad (13)$$

$$L_2\big(Q, y^{(n)}\big) = L_E\big(Q, y^{(n)}\big) + \lambda \, d_{KL}\big(Q, Q_0\big) \qquad (14)$$

$$L_E\big(Q, y^{(n)}\big) = \mathbb{E}_{\boldsymbol{\theta} \sim Q} \, L_1\big(\boldsymbol{\theta}, y^{(n)}\big) \,. \qquad (15)$$

The regularisation parameter $\lambda$ might be needed in the general case where $L_1$ is any loss function not necessarily linked to an underlying density $f$. In that case, because $L_1$ could be scaled differently, the "fidelity-to-data" and "fidelity-to-prior" parts might not be calibrated, and hence need to be recalibrated through $\lambda$ [2].

Assuming that the prior is the same for all data points (hence does not depend on $n$) and that level-2 predictions $Q$ for instances $\boldsymbol{x}$ are of the form $Q = H_{\boldsymbol{\phi}}(\boldsymbol{x})$, where $\boldsymbol{\phi}$ is indexing hypotheses (i.e., the hypothesis space is of the form $\mathcal{H} = \{H_{\boldsymbol{\phi}} \mid \boldsymbol{\phi} \in \Phi\}$) and can be thought of as the model parameters fit to the data $\mathcal{D}$, we obtain

$$L(\boldsymbol{\phi}, \mathcal{D}) = N^{-1} \sum_{n=1}^{N} L_E\big(H_{\boldsymbol{\phi}}\big(\boldsymbol{x}^{(n)}\big), y^{(n)}\big) + \lambda \, d_{KL}\big(H_{\boldsymbol{\phi}}\big(\boldsymbol{x}^{(n)}\big), Q_0\big) \,. \qquad (16)$$

Moreover, if $Q_0$ is the uniform distribution, then the latter is the same as

$$L(\boldsymbol{\phi}, \mathcal{D}) = N^{-1} \sum_{n=1}^{N} L_E\big(H_{\boldsymbol{\phi}}\big(\boldsymbol{x}^{(n)}\big), y^{(n)}\big) - \lambda \operatorname{ENT}\big(H_{\boldsymbol{\phi}}\big(\boldsymbol{x}^{(n)}\big)\big) \,, \qquad (17)$$

since the KL-divergence reduces to the (negative) entropy of the posterior. This is essentially the loss that is also used in the Posterior Network method [3], where $L_1$ in (15) is given by the cross-entropy, and in the evidential networks approach [22], with $L_1$ being the Brier score.

Note that, with a level-2 hypothesis $H_{\boldsymbol{\phi}}$, we can naturally associate the level-1 hypothesis

$$h_{\boldsymbol{\phi}} : \mathcal{X} \longrightarrow \Delta_K, \boldsymbol{x} \mapsto \mathbb{E}_{\boldsymbol{\theta} \sim H_{\boldsymbol{\phi}}(\boldsymbol{x})} \, \boldsymbol{\theta} \,, \qquad (18)$$

which makes point predictions in the form of single probability distributions. For example, if level-2 hypotheses $H_{\boldsymbol{\phi}}$ are Dirichlet, i.e., $H_{\boldsymbol{\phi}}(\boldsymbol{x}) = \operatorname{Dir}(\boldsymbol{\alpha})$, then $h_{\boldsymbol{\phi}}(\boldsymbol{x}) = \big(\theta_1(\boldsymbol{x}), \ldots, \theta_K(\boldsymbol{x})\big)$, where

$$\theta_k(\boldsymbol{x}) = \alpha_k(\boldsymbol{x}) \big/ \sum_{j=1}^{K} \alpha_j(\boldsymbol{x}) \,. \qquad (19)$$

### 3.3  Discussion

The deviation from the prior $Q_0$ obviously serves as a regulariser in (11), but it can also be interpreted from an uncertainty quantification point of view. Recall that (11) can be seen as a compromise between fidelity to data and fidelity to prior. Suppose the learner delivers the level-2 prediction $Q$, knowing that the final (level-1) prediction $\boldsymbol{\theta}$ will be sampled from that distribution, i.e., $\boldsymbol{\theta} \sim Q$. If the learner has a good guess about the true $\boldsymbol{\theta}^*$, it should concentrate $Q$ in the corresponding region in $\Delta_K$, making sure that the sampled $\boldsymbol{\theta}$ will be close to $\boldsymbol{\theta}^*$. To this end, however, it has to deviate from the prior $Q_0$, which causes a cost $d_{KL}(Q, Q_0)$. The latter can thus be seen as a measure of the

---

[2]As a technical assumption, the loss $L_1$ must be such that $0 < \int \exp(-L_1(\boldsymbol{\theta}, y) Q_0(d\boldsymbol{\theta}) < \infty$.

strength of the learner's belief, i.e., the cost it is willing to pay for the concentration, and therefore as a measure of its certainty. This matches quite perfectly with the use of the entropy of $Q$ as a measure of epistemic uncertainty, which is obtained when $Q_0$ is the uniform distribution.

In this regard, it is also worth mentioning that a level-2 prior $Q_0$, i.e., a distribution on $\Delta_K$, is not directly comparable with a level-1 prior $\boldsymbol{\theta}_0$, i.e., a distribution on $\mathcal{Y}$. This is mainly because

- a distribution $Q_0$ represents information about a ground truth $\boldsymbol{\theta}^*$, which, even if unknown (and treated as a random variable by the Bayesian approach), is assumed to exist and to be unique,
- whereas a distribution $\boldsymbol{\theta}_0$ represents information about a random outcome $Y$.

For the purpose of illustration, take again the uniform prior as an example. Taking the uniform prior $\boldsymbol{\theta}_{uni}$ on $\mathcal{Y}$ as a representation of complete ignorance is often criticised for the reason that this representation does not allow for distinguishing between a real lack of knowledge about the ground truth $\boldsymbol{\theta}^*$ and perfectly knowing that $\boldsymbol{\theta}^* = \boldsymbol{\theta}_{uni}$; in the first case, $\boldsymbol{\theta}_{uni}$ is meant to represent the learner's epistemic state, in the latter case it rather refers to an objective reality. On the epistemic level, however, this confusion is actually not possible, because the uniform distribution $Q_{uni}$ on $\Delta_K$ cannot have the second interpretation: Assuming that there is a unique ground truth $\boldsymbol{\theta}^*$, $Q_{uni}$ cannot represent an objective reality. Instead, it is clear that it represents the learner's knowledge.

Note that the prior $Q_0$ in (14) should not be confused with a prior on the hypothesis space $\mathcal{H}$ either. The latter is commonly required in Bayesian learning. Assuming that hypotheses are identified by parameters $\phi \in \Phi$, as we did in our setting, it would be a distribution on the parameter space $\Phi$. Formally, a prior on $\mathcal{H}$ would result in a *single* regularisation term that is added to the entire (cumulative) empirical loss. As opposed to this, $Q_0$ is a *pointwise* penalty, which is part of the level-2 loss function and applied to every training example $(\boldsymbol{x}^{(n)}, y^{(n)})$ individually.

The choice of the uniform distribution as a prior $Q_0$ appears to be natural, especially with the "cost for concentration" interpretation in mind, because the uniform distribution is least concentrated among all distributions. Restrictively, of course, one has to say that a natural uniform prior may not exist in cases where the parameter space $\Theta$ is not bounded.

According to our discussion so far, the loss minimisation approach (14) appears to be quite appealing. However, this approach is not without problems either. First, the loss minimiser $Q$ will depend on the loss function $L_1$, i.e., different representations of uncertainty will be obtained for different losses. This is questionable, because, even if a point prediction — the "action" taken by the learner in the end — will clearly be influenced by the loss, one may wonder whether the representation of uncertainty should not be independent. What this suggests is that (14) incentivises the learner to represent its uncertainty about *the best prediction* rather than *the ground truth*.

As another, possibly more severe problem, note that the incentive to concentrate the prediction $Q$ and to deviate from the prior $Q_0$ will also depend on two other factors: first, the actual ground truth $\boldsymbol{\theta}^*$ itself, and second, the regularisation parameter $\lambda$. For illustration, take again the example of coin tossing. As for the ground truth, note that a deviation from the uniform prior will certainly be beneficial if the coin is very biased ($\theta_1^*$ is close to 0 or 1). In that case, sampling $\theta_1$ uniformly at random will likely end up in a poor prediction. However, if the coin is fair ($\theta_1^* \approx 1/2$), the learner will gain very little by concentrating $Q$ around $\theta_1 = 1/2$, because predicting $\boldsymbol{\theta} = (1/2, 1/2)$ with probability 1 will yield roughly the same expected loss $\mathbb{E}_{y \sim B(\theta_1)} L_1(\boldsymbol{\theta}, y)$ as sampling $\boldsymbol{\theta}$ at random. Thus, there is little incentive for the learner to concentrate in this case, especially considering that a concentration causes a cost.

This brings us to the second factor: the regularisation parameter $\lambda$ has a direct influence on how costly a concentration is. Therefore, it determines the optimal compromise between $L_E$ and $d_{KL}$, and thereby the uncertainty represented by the learner. Again, this is a questionable property, as it renders the uncertainty representation rather arbitrary. An illustration is shown in Fig. 1, where the expectation of loss $L_2(Q, y)$ in (13) with $L_1$ being the cross-entropy loss is plotted for different concentrations, regularisation parameters, and ground-truth parameters. As can be seen by the minima, the optimal concentration depends on both, the regularisation and the ground truth. Thus, even when the learner perfectly knows the ground truth, it may not predict a $Q$ peaked around this parameter, but instead a much flatter distribution, thereby suggesting to be more uncertain than it actually is.

In this regard, we should also put our previous argument in perspective, namely, that a uniform (or, more generally, non-peaked) level-2 distribution cannot represent the ground truth. Although this is in principle true, the task of the learner, as incentivised by the loss function, is not to learn the ground

truth $\boldsymbol{\theta}^*$ in the first place, but rather to make loss-minimising predictions, and nothing prevents it from predicting a flat $Q$ if it serves the purpose of incurring low loss. Also note that the influence of the prior does not diminish with an increasing sample size, like in standard Bayesian learning, because there is one prior per training instance (instead of a single prior for the entire data set).

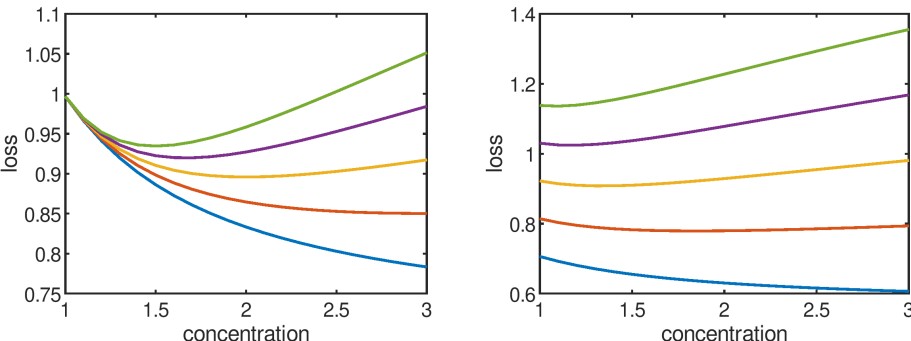

Figure 1: Left: The expectation of the loss $L_2(Q, y)$ in (13) as a function of the concentration $c$ in $Q = \mathrm{Dir}(c, c)$ and for different parameters $\lambda \in \{0, 1/4, 1/2, 3/4, 1\}$ when $L_1$ is the cross-entropy loss and $\boldsymbol{\theta}^* = (1/2, 1/2)$. Right: The same for $\boldsymbol{\theta}^* = (1/4, 3/4)$ and $Q = \mathrm{Dir}(c, 3c)$.

Coming back to our example of tossing a fair coin, a prediction of $Q = Q_{uni}$ should neither be interpreted as "not knowing the ground-truth $\boldsymbol{\theta}^*$", nor as "not knowing the best prediction", but rather as "knowing the prediction does not matter". This is of course also related to and essentially caused by the information gap, i.e., learning a level-2 prediction based on level-0 (instead of level-1) feedback. In fact, if the predictions $\boldsymbol{\theta} \sim Q$ would be compared to observed distributions $\boldsymbol{\theta}^{(n)}$ instead of observed class labels $y^{(n)}$, this would clearly call for concentrating $Q$ around $\boldsymbol{\theta}^*$.

## 4 Formal Results

In the beginning of the previous section, we asked for the existence of a level-2 loss function that incentivises the learner to report epistemic uncertainty in an honest way. Based on our previous discussion, one may wonder whether such a loss can exist. In this section, we provide negative results for both level-2 loss functions considered above. We give these results for the simple coin tossing scenario, but they can be easily extended to more complex settings. What we mean by "appropriate loss" is specified as follows.

**Definition 1.** *A level-2 loss function* $L_2 : \Delta_K^{(2)} \times \mathcal{Y} \longrightarrow \mathbb{R}_+$, *is* appropriate *if the following holds for the empirical loss minimiser* $Q^{(N)} = \mathrm{argmin}_Q \sum_{n=1}^N L_2\left(Q, y^{(n)}\right)$ *on any i.i.d. observational data sequence* $y^{(1)}, y^{(2)}, \dots$ *with* $y^{(i)} \sim \boldsymbol{\theta}^*$:

(A1) *For any sample size* $N$, $\mathbb{E}_{y^{(1:N)}}(U(Q^{(N)})) \geq \mathbb{E}_{y^{(1:N+1)}}(U(Q^{(N+1)}))$, *where* $U$ *is an uncertainty measure*[3] *and* $y^{(1:N)}$ *abbreviates* $y^{(1)}, \dots, y^{(N)}$. *Moreover, there exist some* $\tilde{N}$ *and* $k$ *such that* $\mathbb{E}_{y^{(1:\tilde{N})}}(U(Q^{(\tilde{N})})) > \mathbb{E}_{y^{(1:\tilde{N}+k)}}(U(Q^{(\tilde{N}+k)}))$.

(A2) $Q^{(N)} \xrightarrow{\mathbb{P}} \delta_{\boldsymbol{\theta}^*}$ *as* $N \to \infty$, *where* $\delta_{\boldsymbol{\theta}^*}$ *is the Dirac measure at* $\boldsymbol{\theta}^*$.

In words, $(A1)$ stipulates that the learner's uncertainty should gradually decrease (in expectation) with increasing sample size $N$. In the beginning, the empirical loss minimiser $Q^{(N)}$ should represent high uncertainty (and ideally be uniform for $N = 0$), and the larger $N$, the less uncertain $Q^{(N)}$ should be in terms of the uncertainty measure $U$. $(A2)$ states that in the limit, i.e., if the sample size goes to infinity, all epistemic uncertainty should disappear, and consequently the empirical loss minimiser should converge (in probability) to the Dirac measure $\delta_{\boldsymbol{\theta}^*}$, putting the entire probability mass on the ground-truth $\boldsymbol{\theta}^*$. Both of these assumptions are natural (and minimal) requirements for a suitable level-2 loss function, since honesty is required with respect to the epistemic uncertainty specification on the one side, and consistency of the empirical loss minimiser on the other side.

---

[3]For instance, $U$ might be the entropy ENT.

**Averaging Level-1 Losses.** The following theorem shows that a loss minimisation approach using a level-2 loss (8) with commonly used level-1 losses such as the Brier score or the log-loss (or cross-entropy) does not lead to an appropriate level-2 loss (cf. Section C for the proof).

**Theorem 1.** *For any level-1 loss function $L_1 : \Delta_K \times \mathcal{Y} \longrightarrow \mathbb{R}_+$ that satisfies $L_1\left(\mathbb{E}_{\boldsymbol{\theta} \sim Q}\, \boldsymbol{\theta}, \cdot\right) \leq \mathbb{E}_{\boldsymbol{\theta} \sim Q} L_1\left(\boldsymbol{\theta}, \cdot\right)$, the level-2 loss in (8) is not appropriate.*

Both the Brier score and the log-loss are convex and therefore satisfy the property of Theorem 1. Moreover, the proof reveals an even worse property, namely that the empirical loss minimiser is always a Dirac measure, regardless of the sample size $N$. Consequently, epistemic uncertainty is essentially never reported in a proper way.

**Bayesian Losses and Regularisation.** Next, we show that loss minimisation using Bayesian losses as in (14) does not lead to an appropriate level-2 loss either, if instantiated with commonly used level-1 losses. First, we consider the class of level-1 losses being locally Lipschitz-continuous around the minimiser and strictly proper. Formally, let $d^{(1)}$ be some appropriate metric on $\Delta_K$, then there exists some constant $\mathcal{L} > 0$ depending on $L_1$ such that for any $\boldsymbol{\theta}^* \in \Delta_K$ and any $\boldsymbol{\theta}$ which is close to $\boldsymbol{\theta}^*$ (in a neighborhood $\mathcal{N}(\boldsymbol{\theta}^*)$ in terms of $d^{(1)}$) it holds that

$$\mathbb{E}_{Y \sim \boldsymbol{\theta}^*} L_1(\boldsymbol{\theta}, Y) - \mathbb{E}_{Y \sim \boldsymbol{\theta}^*} L_1(\boldsymbol{\theta}^*, Y) < \mathcal{L}\, d^{(1)}(\boldsymbol{\theta}, \boldsymbol{\theta}^*). \tag{20}$$

The following theorem, the proof of which is deferred to Section D, shows that choosing too large a value for the regularisation parameter $\lambda$ in (14) leads to a violation of Assumption A2.

**Theorem 2.** *For any strictly proper level-1 loss function $L_1$ satisfying (20) for any $\boldsymbol{\theta}, \boldsymbol{\theta}^* \in \Delta_K$, and if $\lambda > 0$ is such that there exists some $\tilde{Q} \in \Delta_K^{(2)}$ with support inside $\mathcal{N}(\boldsymbol{\theta}^*)$, the neighborhood of $\boldsymbol{\theta}^*$, and $\frac{\mathcal{L}\, \tilde{Q}(N(\boldsymbol{\theta}^*))\, \sup_{\boldsymbol{\theta} \in \mathcal{N}(\boldsymbol{\theta}^*)} d^{(1)}(\boldsymbol{\theta}, \boldsymbol{\theta}^*)}{\mathrm{ENT}(\tilde{Q})} < \lambda$, where $\tilde{Q}(N(\boldsymbol{\theta}^*))$ is the probability mass assigned to $\mathcal{N}(\boldsymbol{\theta}^*)$ by $\tilde{Q}$, then the level-2 loss (14) is not appropriate.*

The following theorem shows that choosing the regularisation parameter $\lambda$ in (14) too low leads to a violation of Assumption A1 for level-1 losses similar to those in Theorem 1 (cf. Section E). For this purpose, denote by $\tilde{\Delta}_K^{(2)}$ the subset of $\Delta_K^{(2)}$ consisting of all non-Dirac measures in $\Delta_K^{(2)}$.

**Theorem 3.** *Let $L_1 : \Delta_K \times \mathcal{Y} \longrightarrow \mathbb{R}_+$ be a level-1 loss function such that for any $Q \in \tilde{\Delta}_K^{(2)}$ there exists $\varepsilon_Q > 0$ satisfying $\mathbb{E}_{\boldsymbol{\theta} \sim Q} L_1\left(\boldsymbol{\theta}, \cdot\right) - L_1\left(\mathbb{E}_{\boldsymbol{\theta} \sim Q}\, \boldsymbol{\theta}, \cdot\right) \geq \varepsilon_Q$. Then, the level-2 loss in (14) is not appropriate if $\lambda \leq \inf_{Q \in \tilde{\Delta}_K^{(2)}} \frac{\varepsilon_Q}{\mathrm{ENT}(Q)}$.*

Note that both the Brier score and the log-loss are strictly proper, strictly convex and satisfy the local Lipschitz property (20).

**Discussion.** The above results are general in the sense that $Q$ can be any level-2 distribution, not necessarily restricted to Dirichlet distributions. Moreover, the results do not depend on the underlying uncertainty measure $U$ in Assumption A1 as long as $U$ is not constant as well as maximal for the uniform distribution and minimal for Dirac measures. A key problem of the loss minimisation approach, as revealed by the above results, is the following: The quality (and hence loss) of a prediction $Q$ cannot be judged solely in the context of a single observation $y$. For example, a very uncertain prediction $Q$ (e.g., close to uniform) is completely fine in the beginning, when $N$ is small, but less desirable when $N$ grows large. In principle, the loss should also consider the current knowledge about $\boldsymbol{\theta}^*$, which, however, is missing from its arguments. For the loss (14), this concretely means that the penalty term should be higher in the beginning and lower later on, which could be achieved by specifying $\lambda$ as a decreasing function of $N$ rather than a constant. Then, however, the learner's uncertainty is again controlled in an external way. Due to space limitations, we defer further discussion of the theoretical results to the Appendix F.

## 5 Experiments

In the following, we investigate our findings regarding the empirical loss minimiser (ELM) (see Definition 1) in a simulation study on synthetic data. We consider two representative scenarios for the binary classification setting (i.e., $K = |\mathcal{Y}| = 2$): the scenario with the highest aleatoric uncertainty, where $p(y) = B(0.5)$ and a low aleatoric uncertainty scenario, where $p(y) = B(0.05)$. Note

that the latter is representative of an imbalanced learning scenario. For each scenario, we generate repeatedly observations of different sizes $N$ and compute the corresponding ELM for the Brier score as the underlying level-1 loss function in each run (the results for cross-entropy are similar and hence omitted). As optimizing over all possible level-2 distributions is computationally expensive, we restrict the optimization to two-component mixtures of Dirichlet distributions. In the following table, we report the mean entropy (together with the standard deviations) of the ELM's averaged over 10 runs in dependence on the data set size $N$ for different values of $\lambda$ for both scenarios:

| | $p(y) = B(0.5)$ | | | | | $p(y) = B(0.05)$ | | | | |
|---|---|---|---|---|---|---|---|---|---|---|
| | N = 10 | N = 100 | N = 1000 | N = 10000 | N = 100000 | N = 10 | N = 100 | N = 1000 | N = 10000 | N = 100000 |
| $\lambda = 0$ | 0.000 (0.000) | 0.000 (0.000) | 0.000 (0.000) | 0.000 (0.000) | 0.000 (0.000) | 0.000 (0.000) | 0.000 (0.000) | 0.000 (0.000) | 0.000 (0.000) | 0.000 (0.000) |
| $\lambda = 10^{-5}$ | 0.000 (0.000) | 0.000 (0.000) | 0.000 (0.000) | 0.000 (0.000) | 0.000 (0.000) | 0.000 (0.000) | 0.000 (0.000) | 0.000 (0.000) | 0.000 (0.000) | 0.000 (0.000) |
| $\lambda = 10^{-1}$ | 7.382 (1.278) | 6.530 (0.000) | 4.869 (0.000) | 3.441 (0.734) | 1.575 (0.056) | 7.520 (0.368) | 6.350 (0.250) | 4.851 (0.002) | 3.206 (0.000) | 1.550 (0.046) |
| $\lambda = 0.5$ | 9.216 (0.113) | 7.689 (0.001) | 5.980 (0.158) | 4.369 (0.001) | 2.728 (0.026) | 8.349 (0.174) | 7.148 (0.127) | 5.928 (0.024) | 4.360 (0.032) | 2.707 (0.000) |
| $\lambda = 1$ | 9.615 (0.057) | 8.189 (0.001) | 6.430 (0.316) | 4.935 (0.209) | 3.215 (0.009) | 8.852 (0.224) | 7.598 (0.172) | 6.330 (0.082) | 4.852 (0.002) | 3.207 (0.000) |
| $\lambda = 10$ | 9.958 (0.009) | 9.678 (0.011) | 8.190 (0.000) | 6.530 (0.000) | 4.870 (0.001) | 9.910 (0.012) | 8.969 (0.044) | 7.513 (0.051) | 6.364 (0.016) | 4.851 (0.000) |
| $\bar{\theta}$ | 0.487 (0.108) | 0.514 (0.064) | 0.497 (0.019) | 0.501 (0.005) | 0.500 (0.002) | 0.215 (0.028) | 0.098 (0.013) | 0.055 (0.005) | 0.050 (0.002) | 0.050 (0.001) |

Note that the differential entropy the level-2 distributions takes values in $\mathbb{R}_-$ so for ease of comparison we state here the (Shannon) entropy of the quantized version of the level-2 distributions [4], where a value of zero corresponds to a one-point distribution and the uniform distribution has a value of 9.967 in this case (i.e., 1000 bins are used). Furthermore, the last line reports the empirical mean for the corresponding sample size. We see that for small values of $\lambda$ the ELM is a point-mass (entropy of zero), while for large values of $\lambda$ and huge sample sizes the ELM is still far away from being a point-mass, which is in line with our theoretical results. Moreover, for increasing values of $\lambda$, the ELM changes quite abruptly from one extreme (maximum certainty) to the other (maximum uncertainty) when the sample size is small. Finally, the reported level-2 uncertainty is in both scenarios essentially the same for all different choices of $\lambda$ and sample sizes $N$, although the level-1 uncertainties are quite different. This demonstrates that the influence of $\lambda$ is in a sense quite arbitrary on the faithful epistemic uncertainty representation, as it simply prevents the learner from being too confident without taking the underlying data-generating distribution into account. In Appendix G we also consider a multi-class setting, where similar observations are made.

## 6 Conclusion

In machine learning, it is well known that probabilistic classifiers can be trained by empirical loss minimisation: Suitably chosen loss functions, so-called proper scoring rules, incentivise the learner to predict probabilities in an unbiased way, or, stated differently, the learner minimises expected loss if (and only if) it predicts the true (conditional) probability distribution. In this paper, we investigated the question whether second-order predictors can be trained in a similar way. This is motivated by recent proposals in the literature, where corresponding methods are used to represent the learner's epistemic uncertainty. Obviously, the problem is more challenging, especially due to the lack of an objective ground truth, and because a level-2 representation has to be learned from level-0 data, i.e., data at the observational level where only class labels but no probabilities (level-1 data) are observed. Indeed, our results are negative in the following sense: Level-2 loss functions do not incentivise the learner to predict its epistemic uncertainty in a faithful way. Instead, to minimise the loss in expectation, the learner is encouraged to pretend more (or less) confidence than warranted. Besides, contrary to what one would expect, the uncertainty does not decrease with an increasing sample size. Thus, we believe that the empirical findings reported in the literature should be reconsidered and carefully analyzed in light of our results. While it is true that good results are obtained in the majority of existing work, it is also true that the reasons for these results are not always fully transparent. Our results confirm the difficulty of epistemic uncertainty quantification, which, we believe, is a general problem that also applies to other approaches. Strictly speaking, however, our formal results only hold for the loss functions that have been proposed in the literature so far, and hence do not completely exclude the existence of other types of losses providing the right incentives for the learner. As future work, we plan to further elaborate on this problem, either coming up with an appropriate loss function or proving that such a loss cannot exist.

## Acknowledgments and Disclosure of Funding

Willem Wageman received funding from the Flemish Government under the "Onderzoeksprogramma Artificiële Intelligentie (AI) Vlaanderen" Programme.

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
