# A List of Symbols

The following table contains a list of symbols that are frequently used in the main paper as well as in the following supplementary material.

| | |
|---|---|
| **General Learning Setting** | |
| $K$ | number of classes |
| $\mathcal{X}$ | instance space |
| $\mathcal{Y}$ | label space with labels $y_1, \ldots, y_K$ |
| $\mathcal{D}$ | training data $\left\{ \left( \boldsymbol{x}^{(n)}, y^{(n)} \right) \right\}_{n=1}^{N} \subset \mathcal{X} \times \mathcal{Y}$ |
| $P$ | data generating probability |
| $p(\cdot \,\vert\, \boldsymbol{x})$ | conditional distribution on $\mathcal{Y}$, i.e., $p(y \,\vert\, \boldsymbol{x})$ probability to observe $y$ given $\boldsymbol{x}$ |
| $\mathbb{P}(\mathcal{Y})$ | the set of probability distributions on $\mathcal{Y}$ |
| $\Delta_K$ | the $K$-simplex, i.e., $\Delta_K := \left\{ \boldsymbol{\theta} = (\theta_1, \ldots, \theta_K) \in [0,1]^K \,\vert\, \|\boldsymbol{\theta}\|_1 = 1 \right\}$ |
| $\boldsymbol{\theta} = (\theta_1, \ldots, \theta_K)^\top$ | probability vector with K atoms, i.e., an element of $\Delta_K$ |
| $\boldsymbol{\theta}_{uni} := (1/K, \ldots, 1/K)^\top$ | uniform distribution on $\mathcal{Y}$ (an element of $\Delta_K$) |
| $\boldsymbol{\theta}^* = \boldsymbol{\theta}^*(\boldsymbol{x})$ | (conditional) distribution on $\mathcal{Y}$, i.e., the ground-truth |
| **Level-1 Learning Setting** | |
| $\mathcal{H}$ | (level-1) hypothesis space consisting of hypothesis $h : \mathcal{X} \longrightarrow \Delta_K$ |
| $L_1$ | loss function for level-1 hypothesis, i.e., $L_1 : \Delta_K \times \mathcal{Y} \longrightarrow \mathbb{R}$ |
| $R_{emp}(\cdot)$ | empirical loss of a level-1 hypothesis (cf. (3)) |
| $R(\cdot)$ | risk or expected loss of a level-1 hypothesis (cf. (2)) |
| $\hat{h}$ | empirical risk minimiser, i.e., $\hat{h} = \operatorname{argmin}_{h \in \mathcal{H}} R_{emp}(h)$ |
| $h^*$ | true risk minimiser or Bayes predictor, i.e., $h^* = \operatorname{argmin}_{h \in \mathcal{H}} R(h)$ |
| **Level-2 Learning Setting** | |
| $\Delta_K^{(2)}$ | the set of distributions on $\Delta_K$ |
| $H$ | (level-2) hypothesis, i.e., a mapping $h : \mathcal{X} \longrightarrow \Delta_K^{(2)}$ |
| $H_\phi$ | indexed (level-2) hypothesis, where $\phi$ is an indexing hypothesis |
| $h_\phi$ | level-1 hypothesis induced by $H_\phi$ (cf. (18)) |
| $Q$ | probability distribution on $\Delta_K$, i.e., an element of $\Delta_K^{(2)}$ |
| $Q_{uni}$ | uniform distribution on $\Delta_K$ (an element of $\Delta_K^{(2)}$) |
| $L_2$ | loss function for level-2 hypothesis, i.e., $L_2 : \Delta_K^{(2)} \times \mathcal{Y} \longrightarrow \mathbb{R}_+$ |
| $L_E$ | expected level-1 loss (cf. (15)) |
| $\lambda$ | regularisation parameter (cf. (14)) |
| $R_{emp}^{(2)}(\cdot)$ | empirical (level-2) loss of a level-2 hypothesis (appears only in the appendix) |
| $R^{(2)}(\cdot)$ | (level-2) risk or expected loss of a level-2 hypothesis (appears only in the appendix) |
| $Q^{(N)}$ | empirical level-2 risk minimiser (for coin tossing problem), i.e., $Q^{(N)} = \operatorname{argmin}_{Q \in \Delta_K^{(2)}} R_{emp}^{(2)}(Q)$ |
| **Distributions** | |
| $B(\theta)$ | Bernoulli distribution with parameter $\theta \in [0,1]$ |
| $\mathrm{Cat}(\boldsymbol{\theta})$ | Categorical distribution with parameter $\boldsymbol{\theta} \in \Delta_K$ |
| $\mathrm{Dir}(\boldsymbol{\alpha})$ | Dirichlet distribution with parameter $\boldsymbol{\alpha} \in \mathbb{R}_+^K$ |
| $\delta_{\boldsymbol{\theta}}$ | Dirac measure at $\boldsymbol{\theta} \in \Delta_K$ |
| $\xrightarrow{\mathbb{P}}$ | convergence in distribution |
| **Entropy and Divergence** | |
| $\mathrm{ENT}(\cdot)$ | Shannon entropy (on $\Delta_K^{(2)}$) |
| $d_{KL}(\cdot, \cdot)$ | Kullback-Leibler divergence (on $\Delta_K^{(2)} \times \Delta_K^{(2)}$) |
| $d^{(1)}(\cdot, \cdot)$ | some metric on $\Delta_K$ |
| $U(\cdot)$ | an uncertainty measure (on $\Delta_K^{(2)}$), see Definition 1 |

# B The Dirichlet Distribution

A Dirichlet distribution $\mathrm{Dir}(\alpha)$ is specified by means of $K \geq 2$ positive real-valued parameters, i.e., a vector $\boldsymbol{\alpha} = (\alpha_1, \ldots, \alpha_K) \in \mathbb{R}_+^K$. The probability density function is defined on the $K$ simplex

$$\Delta_K = \left\{ \boldsymbol{\theta} = (\theta_1, \ldots, \theta_K)^\top \mid \theta_1, \ldots, \theta_K \geq 0, \sum_{k=1}^K \theta_k = 1 \right\}$$

and given as follows:

$$p(\boldsymbol{\theta} \mid \boldsymbol{\alpha}) = p(\theta_1, \ldots, \theta_K \mid \boldsymbol{\alpha}) = \frac{1}{\mathbb{B}(\boldsymbol{\alpha})} \prod_{k=1}^K \theta_k^{\alpha_k - 1},$$

where the normalisation constant is the multivariate beta function:

$$\mathbb{B}(\boldsymbol{\alpha}) = \frac{\prod_{k=1}^K \Gamma(\alpha_k)}{\Gamma\left(\sum_{k=1}^K \alpha_k\right)},$$

with $\Gamma$ denoting the gamma function. In Bayesian statistics, the Dirichlet distribution is commonly used as the conjugate prior of the multinomial distribution. From a machine learning perspective, this makes it quite attractive for the (multi-class) classification setting.

The parameters $\alpha_k$ can be interpreted as *evidence* in favour of the $k^{th}$ category: the larger $\alpha_k$, the larger the probability for a high $\theta_k$, and hence the higher the probability to observe the $k^{th}$ category as outcome. More specifically, the expected value of $\theta_k$ (and hence the natural estimate $\hat{\theta}_k$) is given by

$$\mathbb{E}(\theta_k) = \frac{\alpha_k}{\sum_{j=1}^K \alpha_j}.$$

Moreover, the larger the parameters $\alpha_k$ and hence the sum $\alpha_0 = \sum_{j=1}^K \alpha_j$, the more "peaked" the Dirichlet distribution becomes. For $\alpha_1 = \ldots = \alpha_K = 1$, the uniform distribution on $\Theta$ is obtained, i.e., the "least informed" distribution with highest entropy. For $\alpha_1 = \ldots = \alpha_K = c$, with $c$ the so-called concentration parameter, the distribution on $\Theta$ remains symmetric. However, while it peaks at $\boldsymbol{\theta} = (1/K, \ldots, 1/K)$ for larger $c > 1$, it becomes more dispersed and assigns higher probability mass around the "corners" of the probability simplex ($\theta_k = 1$ and $\theta_j = 0$ for all $j \neq k$) for $c$ close to 0.

As already said, the Dirichlet distribution is conjugate to the multinomial distribution. More specifically, Bayesian updating of a prior $\mathrm{Dir}(\alpha_1, \ldots, \alpha_K)$ in light of observed frequencies $c_1, \ldots, c_K$ of the $K$ categories yields the posterior $\mathrm{Dir}(\alpha_1 + c_1, \ldots, \alpha_K + c_K)$. In other words, Bayesian inference comes down to simple counting, which makes it extremely simple. In this regard, the $\alpha_k$ are often interpreted as "pseudocounts" of the categories.

## B.1 Quantifying Epistemic Uncertainty

Suppose that epistemic uncertainty of the learner is represented by means of a Dirichlet $\mathrm{Dir}(\alpha)$. Often, one is interested in quantifying this uncertainty in terms of a single number. What is sought, therefore, is an uncertainty measure $U$ mapping distributions to real numbers. In the literature, various examples of such measures are known, with Shannon entropy the arguably most prominent one. Like Shannon entropy, uncertainty measures are typically derived on an axiomatic basis, i.e., a reasonable measure of uncertainty should obey certain properties [14].

The (differential) entropy of a $\mathrm{Dir}(\boldsymbol{\alpha})$ distribution is given by

$$\mathrm{ENT}(\mathrm{Dir}(\boldsymbol{\alpha})) = \log \mathbb{B}(\boldsymbol{\alpha}) + (\alpha_0 - K)\varphi(\alpha_0) - \sum_{j=1}^K (\alpha_j - 1)\varphi(\alpha_j), \tag{21}$$

where $\varphi$ is the digamma function.

## C  Proof of Theorem 1

Let $R^{(2)}_{emp}(Q) = \frac{1}{N} \sum_{n=1}^{N} L_2\left(Q, y^{(n)}\right)$ be the empirical risk of a level-2 prediction $Q \in \Delta_K^{(2)}$. As we consider a level-2 loss as in (8), the empirical risk is given by

$$R^{(2)}_{emp}(Q) = \frac{1}{N} \sum_{n=1}^{N} \mathbb{E}_{\boldsymbol{\theta} \sim Q} L_1\left(\boldsymbol{\theta}, y^{(n)}\right).$$

By assumption on the level-1 loss $L_1$, it holds that

$$R^{(2)}_{emp}(Q) \geq \frac{1}{N} \sum_{n=1}^{N} L_1\left(\mathbb{E}_{\boldsymbol{\theta} \sim Q}\, \boldsymbol{\theta}, y^{(n)}\right).$$

Let $\widetilde{Q}^{(N)}$ be the minimiser over all $Q \in \Delta_K^{(2)}$ of the right-hand side, then $\tilde{\boldsymbol{\theta}}^{(N)} = \mathbb{E}_{\boldsymbol{\theta} \sim \widetilde{Q}^{(N)}} \boldsymbol{\theta}$ is an element in $\Delta_K$. Define $\hat{Q}^{(N)} = \delta_{\tilde{\boldsymbol{\theta}}^{(N)}}$ and note that $\mathbb{E}_{\boldsymbol{\theta} \sim \hat{Q}^{(N)}} \boldsymbol{\theta} = \tilde{\boldsymbol{\theta}}^{(N)}$. Then,

$$
\begin{aligned}
R^{(2)}_{emp}(\hat{Q}^{(N)}) &= \frac{1}{N} \sum_{n=1}^{N} \mathbb{E}_{\boldsymbol{\theta} \sim \hat{Q}^{(N)}} L_1\left(\boldsymbol{\theta}, y^{(n)}\right) \\
&= \frac{1}{N} \sum_{n=1}^{N} L_1\left(\tilde{\boldsymbol{\theta}}^{(N)}, y^{(n)}\right) \\
&= \frac{1}{N} \sum_{n=1}^{N} L_1\left(\mathbb{E}_{\boldsymbol{\theta} \sim \widetilde{Q}^{(N)}}\, \boldsymbol{\theta}, y^{(n)}\right).
\end{aligned}
$$

Thus, $R^{(2)}_{emp}(Q) \geq R^{(2)}_{emp}(\hat{Q}^{(N)})$ for all $Q \in \Delta_K^{(2)}$. In particular, for any $N$ the empirical loss minimiser is $Q^{(N)} = \hat{Q}^{(N)} = \delta_{\tilde{\boldsymbol{\theta}}^{(N)}}$, so that Assumption A1 is violated.

## D  Proof of Theorem 2

Let $R^{(2)}(Q) = \mathbb{E}_{Y \sim \boldsymbol{\theta}^*} L_2\left(Q, Y\right)$ be the true risk of a level-2 prediction $Q \in \Delta_K^{(2)}$. As $L_2$ is of the form as in (14), the true risk is due to Fubini-Tonelli's theorem given by

$$
\begin{aligned}
R^{(2)}(Q) &= \mathbb{E}_{Y \sim \boldsymbol{\theta}^*} \mathbb{E}_{\boldsymbol{\theta} \sim Q} L_1\left(\boldsymbol{\theta}, Y\right) - \lambda \text{ENT}(Q) \\
&= \mathbb{E}_{\boldsymbol{\theta} \sim Q} \mathbb{E}_{Y \sim \boldsymbol{\theta}^*} L_1\left(\boldsymbol{\theta}, Y\right) - \lambda \text{ENT}(Q).
\end{aligned}
$$

Thus, $R^{(2)}(\delta_{\boldsymbol{\theta}^*}) = \mathbb{E}_{Y \sim \boldsymbol{\theta}^*} L_1\left(\boldsymbol{\theta}^*, Y\right)$, since $\text{ENT}(\delta_{\boldsymbol{\theta}}) = 0$ for any $\boldsymbol{\theta} \in \Delta_K$. Hence, for $\tilde{Q} \in \Delta_K^{(2)}$ such that

$$\frac{\mathcal{L}\, \tilde{Q}(N(\boldsymbol{\theta}^*)) \sup_{\boldsymbol{\theta} \in N(\boldsymbol{\theta}^*)} d^{(1)}(\boldsymbol{\theta}, \boldsymbol{\theta}^*)}{\text{ENT}(\tilde{Q})} < \lambda$$

holds,

$$
\begin{aligned}
R^{(2)}(\tilde{Q}) &= \mathbb{E}_{\boldsymbol{\theta} \sim \tilde{Q}} \mathbb{E}_{Y \sim \boldsymbol{\theta}^*} L_1\left(\boldsymbol{\theta}, Y\right) - \mathbb{E}_{Y \sim \boldsymbol{\theta}^*} L_1\left(\boldsymbol{\theta}^*, Y\right) - \lambda \text{ENT}(\tilde{Q}) + R^{(2)}(\delta_{\boldsymbol{\theta}^*}) \\
&< \mathcal{L}\, \tilde{Q}(N(\boldsymbol{\theta}^*)) \sup_{\boldsymbol{\theta} \in N(\boldsymbol{\theta}^*)} d^{(1)}(\boldsymbol{\theta}, \boldsymbol{\theta}^*) - \lambda \text{ENT}(\tilde{Q}) + R^{(2)}(\delta_{\boldsymbol{\theta}^*}) \\
&< R^{(2)}(\delta_{\boldsymbol{\theta}^*}).
\end{aligned}
$$

Consequently, the true risk minimiser differs from $\delta_{\boldsymbol{\theta}^*}$. Since $L_1$ is a strictly proper loss, Theorem 5.7 by [24] lets us infer that the empirical risk minimiser $Q^{(N)}$ converges in probability to the minimiser of the true risk, which violates Assumption A2.

## E  Proof of Theorem 3

In the following, we abbreviate $L_2\left(Q, y^{(n)}\right)$ by $L_2^{(n)}(Q)$ and $L_1\left(\boldsymbol{\theta}, y^{(n)}\right)$ by $L_1^{(n)}(\boldsymbol{\theta})$. For any $N$, let

$$\widetilde{Q}^{(N)} = \operatorname*{argmin}_{Q \in \tilde{\Delta}_K^{(2)}} \frac{1}{N} \sum_{n=1}^{N} L_2^{(n)}(Q) - \lambda \text{ENT}(Q)$$

and $\tilde{\boldsymbol{\theta}}^{(N)} = \mathbb{E}_{\boldsymbol{\theta} \sim \widetilde{Q}^{(N)}} \, \boldsymbol{\theta}$. Further, set $\hat{Q}^{(N)} = \delta_{\tilde{\boldsymbol{\theta}}^{(N)}}$ and note that

$$R_{emp}^{(2)}(\hat{Q}^{(N)}) = \frac{1}{N} \sum_{n=1}^{N} L_2^{(n)} \left( \hat{Q}^{(N)} \right)$$

$$= \frac{1}{N} \sum_{n=1}^{N} L_1^{(n)} \left( \tilde{\boldsymbol{\theta}}^{(N)} \right)$$

$$= \frac{1}{N} \sum_{n=1}^{N} L_1^{(n)} \left( \mathbb{E}_{\boldsymbol{\theta} \sim \widetilde{Q}^{(N)}} \, \boldsymbol{\theta} \right).$$

With this, we can infer for any $Q \in \tilde{\Delta}_K^{(2)}$ that

$$R_{emp}^{(2)}(Q) = \frac{1}{N} \sum_{n=1}^{N} L_2^{(n)}(Q) - \lambda \mathrm{ENT}(Q)$$

$$= \frac{1}{N} \sum_{n=1}^{N} L_2^{(n)}(Q) - L_1^{(n)} \left( \mathbb{E}_{\boldsymbol{\theta} \sim \widetilde{Q}^{(N)}} \, \boldsymbol{\theta} \right) - \lambda \mathrm{ENT}(Q) + R_{emp}^{(2)}(\hat{Q}^{(N)})$$

$$\geq \frac{1}{N} \sum_{n=1}^{N} L_2^{(n)} \left( \widetilde{Q}^{(N)} \right) - L_1^{(n)} \left( \mathbb{E}_{\boldsymbol{\theta} \sim \widetilde{Q}^{(N)}} \, \boldsymbol{\theta} \right) - \lambda \mathrm{ENT}(\widetilde{Q}^{(N)}) + R_{emp}^{(2)}(\hat{Q}^{(N)})$$

$$\geq \frac{1}{N} \sum_{n=1}^{N} \varepsilon_{\widetilde{Q}^{(N)}} - \lambda \mathrm{ENT}(\widetilde{Q}^{(N)}) + R_{emp}^{(2)}(\hat{Q}^{(N)})$$

$$\geq R_{emp}^{(2)}(\hat{Q}^{(N)}),$$

where the first inequality is by choice of $\widetilde{Q}^{(N)}$, the second last by the assumption on $L_1$, and the last inequality is by choice of $\lambda$. Thus, the empirical loss minimiser is a Dirac measure, regardless of $N$, so that Assumption A1 is violated.

## F   Further Discussion on Theorems 2 and 3

Note that the two ranges for $\lambda$ in Theorems 2 and 3 do not necessarily represent a partition of the positive real numbers, so it would be possible in principle that there exists a range of $\lambda$ values "in between" where Theorems 2 and 3 do not apply. However, one must still note that the respective bounds for the ranges can potentially be brought closer together, as they are chosen rather to simplify the proofs. For example, the choice of the bound for $\lambda$ in Theorem 3 is extreme in the sense that the empirical loss minimiser is always a Dirac measure. By slightly loosening this bound, one could show that the empirical loss minimiser is "almost" a Dirac measure, which, however, would still violate Assumption A1. Similarly the enumerator for $\lambda$ in Theorem 2 is a rather rough estimate due to the supremum and could be tightened by $\mathcal{L} \, \mathbb{E}_{\boldsymbol{\theta} \sim \tilde{Q}}(d^{(1)}(\boldsymbol{\theta}, \boldsymbol{\theta}^*))$. Finally, note that both the Brier score and the log-loss are strictly convex and therefore satisfy the property of Theorem 3 due to Jensen's (strict) inequality.

## G   Further Experiments

In this section, we extend the simulation study from Section 5 regarding the behavior of the empirical loss minimiser (ELM) (see Definition 1) over two-component mixtures of Dirichlet distributions to the multi-class classification setting. Again, we shall resort to synthetic data and two representative scenarios for the multi-class classification setting with three classes: the scenario with the highest aleatoric uncertainty, where

$$p(\cdot) = \mathrm{Cat}\,(1/3, 1/3, 1/3)$$

and a low aleatoric uncertainty scenario, where

$$p(\cdot) = \mathrm{Cat}\,(7/8, 1/16, 1/16)\,.$$

Note that the latter is representative of an imbalanced learning scenario. Following the same procedure as in Section 5, we obtain for the mean entropy (together with the standard deviations) of the ELM's averaged over 10 runs in dependence on the data set size $N$ for different values of $\lambda$ for both scenarios:

| | $p(y) = \text{Cat}(1/3, 1/3, 1/3)$ | | | | | $p(y) = \text{Cat}(7/8, 1/16, 1/16)$ | | | | |
|---|---|---|---|---|---|---|---|---|---|---|
| | N = 10 | N = 100 | N = 1000 | N = 10000 | N = 100000 | N = 10 | N = 100 | N = 1000 | N = 10000 | N = 100000 |
| $\lambda = 0$ | 0.000 (0.000) | 0.000 (0.000) | 0.000 (0.000) | 0.000 (0.000) | 0.000 (0.000) | 0.000 (0.000) | 0.000 (0.000) | 0.000 (0.000) | 0.000 (0.000) | 0.000 (0.000) |
| $\lambda = 10^{-5}$ | 0.000 (0.000) | 0.000 (0.000) | 0.000 (0.000) | 0.000 (0.000) | 0.000 (0.000) | 0.000 (0.000) | 0.000 (0.000) | 0.000 (0.000) | 0.000 (0.000) | 0.000 (0.000) |
| $\lambda = 10$ | 10.174 (0.014) | 9.687 (0.013) | 6.912 (0.002) | 3.621 (0.001) | 1.401 (0.002) | 10.156 (0.139) | 8.059 (0.112) | 5.805 (0.092) | 3.312 (0.021) | 1.201 (0.001) |
| $\hat{\theta}_1$ | 0.321 (0.117) | 0.328 (0.046) | 0.336 (0.017) | 0.332 (0.007) | 0.333 (0.002) | 0.726 (0.088) | 0.850 (0.022) | 0.874 (0.011) | 0.880 (0.004) | 0.875 (0.003) |

For comparison purposes, we report here again the (Shannon) entropies of the quantized version of the level-2 distributions instead of their differential entropies (see Section 5). Since we use 1326 bins, the uniform distribution (on level-2) has an entropy of 10.3729. Thus, the results are consistent with the empirical results for the binary classification setting in Section 5 and, more importantly, with our theoretical results