# OpenReview forum: "Pitfalls of Epistemic Uncertainty Quantification through Loss Minimisation"
_NeurIPS.cc/2022/Conference — NeurIPS 2022 Accept_

### Official Review · Reviewer_Tn2k · 2022-07-11

**Rating:** 7
**Confidence:** 4
**Soundness:** 3 good
**Presentation:** 4 excellent
**Contribution:** 3 good

**Summary:**

This paper discusses the difficulty in learning a predictor that exhibits the epistemic uncertainty given labeled examples as a training dataset.
This is mainly due to the lack of ground truth of epistemic uncertainty in the training set.
Loss functions to handle the uncertainty are apt to result in an arbitrary amount of uncertainty in the prediction.


**Questions:**

1. Can ensemble methods be regarded as a form of averaging level-1 loss with $Q$ being a set of ensemble members in Eq. (8)?
An example of ensembles is particle variational inference.
A. Saeedi et al. "Variational Particle Approximations" JMLR 18(69):1--29, 2017.

2. An exact quantification of epistemic uncertainty may be ill-posed, but having the order of amount in the uncertainty can be useful for making a decision from the prediction.
For example, the uncertainty in $p(y|x_1)$ is greater than that in $p(y|x_2)$.
Is it still difficult to determine the order of uncertainties due to the lack of ground truth, or any of losses discussed in the paper may handle such relationship of magnitude?


**Limitations:**

The theme of this paper is the limitation of epistemic uncertainty quantification.


**Strengths And Weaknesses:**

The strengths of this paper are:
* The topic is relevant and important in the literature.
* The paper is well-organized and easy to follow with many examples.

The weakness is
* a limited amount of numerical results may restrict the understanding of readers.
For example, giving a prediction with various values of $\lambda$ in (15) may improve the clarity even better.

---

> ### Author Response · Authors · 2022-08-02
> **Response to Reviewer Tn2k**
>
> First of all, thank you very much for appreciating the quality and originality of our paper as well as the significance of our results. In the following, we would like to address some of your concerns as well as the questions you raised.
>
> ### Weakness: The weakness is a limited amount of numerical results may restrict the understanding of readers. For example, giving a prediction with various values of $\lambda$ in (15) may improve the clarity even better;
>
> Due to page restrictions, we have given only one experiment for illustration. However, as the accepted papers are allowed an additional content page, it would be no problem to extend the experiments as the reviewer suggests. Thanks for the suggestion. We will expand the paper to this end.
>
>
> ### Question 1. Can ensemble methods be regarded as a form of averaging level-1 loss with Q being a set of ensemble members in Eq. (8)? An example of ensembles is particle variational inference. A. Saeedi et al. "Variational Particle Approximations" JMLR 18(69):1-29, 2017.
>
> Not exactly in our sense, as the ensemble members are level-1 representations, each obtained by using a suitable level-1 loss. Even when using for $Q$ the uniform distribution over the ensemble predictions, this would not correspond to Eq. (8), as there we average over the level-1 losses and not over their predictions.
>
>
>
> ### Question 2. An exact quantification of epistemic uncertainty may be ill-posed, but having the order of amount in the uncertainty can be useful for making a decision from the prediction. For example, the uncertainty in $p(y | x_1) $is greater than that in $p(y |  x_2)$. Is it still difficult to determine the order of uncertainties due to the lack of ground truth, or any of losses discussed in the paper may handle such relationship of magnitude?
>
> The level-2 losses obtained by averaging level-1 losses, with level-1 losses as in Theorem 1, are not suitable for obtaining such an ordering. The reason is that the empirical risk minimiser is a Dirac measure and consequently the ordering is meaningless. A similar issue arises for the Bayesian losses (see Theorem 3), if the regularisation parameter is chosen too low, while choosing the regularisation parameter too high, the level-2 is concentrated in the limit at a wrong level-1 distribution.
>
> More generally, we are not entirely sure that relative probabilities will suffice in the context of decision making. This might be the case if a decision maker has the choice between deciding in situation $x_1$ and deciding in situation $x_2$. However, the more common situation is that a decision must be taken (or rejected) in a given situation $x$. Then, having an absolute notion of uncertainty is really essential. Imagine, for example, an autonomous driving agent, which needs to decide whether to take action or transfer control to the human driver. For this agent, it’s not enough to know that the uncertainty is less than it was in another traffic situation. Instead, it needs to quantify uncertainty in an absolute manner.

---

> > ### Comment · Reviewer_Tn2k · 2022-08-10
> > **Response to feedback**
> >
> > Thank you for the clarification to my questions. The response was very informative.
> >
> > > Due to page restrictions, we have given only one experiment for illustration. However, as the accepted papers are allowed an additional content page, it would be no problem to extend the experiments as the reviewer suggests. Thanks for the suggestion. We will expand the paper to this end.
> >
> > Would be good to see additional results for better significance of the paper.

---

### Official Review · Reviewer_48Wj · 2022-07-11

**Rating:** 6
**Confidence:** 3
**Soundness:** 3 good
**Presentation:** 3 good
**Contribution:** 3 good

**Summary:**

The paper analyzes the recent line of work on uncertainty estimation in neural networks which model distributions over distributions by loss minimization. The authors argue that this paradigm of learning higher order distributions with loss minimization does not lead to meaningful estimates of epistemic uncertainty. The authors support their arguments with theoretical results showing that predictors learned with loss minimization.

**Questions:**

* The results consider methods which learn second-order predictors through loss minimization relying on the input data. But there have been methods proposed in literature [1], which learns a second order predictor relying on features derived from the dataset. How do the theoretical results in this work relate to such approaches?

[1] Jain, M., Lahlou, S., Nekoei, H., Butoi, V., Bertin, P., Rector-Brooks, J., ... & Bengio, Y. (2021). Deup: Direct epistemic uncertainty prediction. arXiv preprint arXiv:2102.08501.

**Limitations:**

* In summary, I think the main limitation of the work is that it is fairly narrow in scope. I believe it is important to critically analyze existing work, and re-examine the validity of approaches, but the impact is limited since the results are restricted to existing losses and the analysis does not reveal any potential fixes.
* Another drawback of the analysis is that it does not account for the good empirical performance of the methods in discussion. This could certainty be attributed in part to empirical evaluation metrics but this gap is something which should be discussed.


**Strengths And Weaknesses:**

**Strengths**
* I believe the paper is quite relevant to the community as it examines a popular class of methods for uncertainty estimation. The theoretical results are an important (albeit limited in scope) contribution to the literature.
* The paper is generally well written. The authors convey their arguments systematically supplemented with enough running examples to follow the argument. The paper is also relatively self-sufficient with all the relevant quantities defined properly. The results and accompanying assumptions are stated clearly.

**Weaknesses**
* The analysis is limited in scope, applying only to a certain losses proposed in prior work. It is not reveal a general negative result and neither does it reveal potential remedies. I believe this limits the impact of the work substantially.
* Another aspect which the analysis does not address is the good empirical performance of the analyzed family of methods. Contrary to the results presented here, these methods exhibit strong empirical performance on various tasks evaluating uncertainty estimation. This gap between the theory and empirical performance in critical and should be discussed.


--- Post Discussion Update ----
The response from the authors addressed some of my concerns. I have updated the score to reflect that.

---

> ### Author Response · Authors · 2022-08-02
> **Response to Reviewer 48Wj**
>
> First of all, thank you very much for appreciating the quality and originality of our paper as well as the significance of our results. In the following, we would like to address some of your concerns and the questions you raised.
>
> ### Weakness 1: The analysis is limited in scope, applying only to a certain losses proposed in prior work. It is not reveal a general negative result and neither does it reveal potential remedies. I believe this limits the impact of the work substantially.
>
> We essentially agree and make this point very explicit (see our abstract, introduction and conclusion). Nevertheless, we believe that our contribution is important, because the usage of the criticized level-2 ERM approaches is rapidly increasing. Accordingly, there should be more clarity on the theoretical level as to what the limitations or pitfalls of these approaches are. Also, one should acknowledge that the classes of loss functions we consider are rather broad and can be instantiated in many ways. Last but not least,  note that our discussion provides general insights that may provide the basis for extending the results beyond these types of losses.
>
> ### Weakness 2: Another aspect which the analysis does not address is the good empirical performance of the analyzed family of methods. Contrary to the results presented here, these methods exhibit strong empirical performance on various tasks evaluating uncertainty estimation. This gap between the theory and empirical performance in critical and should be discussed.
>
>
> Good point, which is clearly worth mentioning and should be addressed in future work. More specifically, we believe that the empirical findings reported in the literature should be reconsidered and carefully analyzed in light of our results. It’s true that strong performance is achieved in several papers, but it’s also true that the reasons for these results are not always fully transparent. We’ll mention this, thanks again.
>
> ### Questions: The results consider methods which learn second-order predictors through loss minimization relying on the input data. But there have been methods proposed in literature [1], which learns a second order predictor relying on features derived from the dataset. How do the theoretical results in this work relate to such approaches?
>
> As mentioned by the reviewer, this is a different approach. In particular, it does *not* use a second-order predictor as a way to represent epistemic uncertainty. Instead, epistemic uncertainty is quantified by the difference between total uncertainty (the expected loss of a given predictor) and the aleatoric uncertainty (error of the Bayes predictor).
> In particular, epistemic uncertainty is reported by means of a function of the features and not by means of a probability over (cond.) distributions, which is the second-order representation we (or rather the criticized approaches) consider.
> Finally, the DEUP approach includes two predictors, one usual predictor (either level-0 or level-1), whose epistemic uncertainty should be reported, and another predictor, which predicts the epistemic uncertainty of the first. Thus, the DEUP approach and the level-2 ERM approaches are in some way complementary. Nevertheless, we will include this approach in our literature review and are thankful for pointing it out.

---

> > ### Comment · Reviewer_48Wj · 2022-08-03
> > **Response to rebuttal**
> >
> > Thank you for the response!
> >
> > > More specifically, we believe that the empirical findings reported in the literature should be reconsidered and carefully analyzed in light of our results. It’s true that strong performance is achieved in several papers, but it’s also true that the reasons for these results are not always fully transparent. We’ll mention this, thanks again.
> >
> > Thank you. I agree that empirical findings in the literature should be reconsidered given the results presented in this work, and there is work to be done in reconciling the differences.
> >
> > > Thus, the DEUP approach and the level-2 ERM approaches are in some way complementary. Nevertheless, we will include this approach in our literature review and are thankful for pointing it out.
> >
> > Thank you for the clarification!
> >
> > I believe the author response as well as comments from other reviewers address the major concerns I had about the scope and limitations of the work. I have increased my score to reflect that.

---

### Official Review · Reviewer_MUhd · 2022-07-12

**Rating:** 7
**Confidence:** 3
**Soundness:** 3 good
**Presentation:** 3 good
**Contribution:** 3 good

**Summary:**

This paper demonstrates the problem of current approaches while modelling epistemic uncertainty by minimising a given loss function.

The discussion is built upon two common approaches to training a model with epistemic (level-2) uncertainty: (1) minimising averaged prediction loss with random samples from the epistemic uncertainty model. (2) the Bayesian approach, where a given prior distribution controls the epistemic uncertainty.

The authors proposed two requirements for the desired loss function for epistemic (level-2) uncertainty and then theoretically demonstrated that both (1) and (2) above cannot meet the requirements under some circumstances.


**Questions:**

While the discussion on the (Average Level-1 Loss) is clear, the reviewer found the debate on (Bayesian Losses and Regulations) requires an additional assumption that the uniform posterior gives the maximal uncertainty. I wonder if the authors can comment if such an assumption is too restrictive?

(1) In a regression setting where the posterior has infinite support, it might be hard to come up with such a uniform distribution without additional transformation.

(2) Also, for a highly unbalanced classification problem, the uniform distribution might not be the case for maximal uncertainty.


========================================

The reviewer found the additional examples and discussion from the authors informative and provided some further justification for the paper. This paper should lead to some nice discussion at the conference.

**Limitations:**

Yes, the author discussed the limitations at the end of the paper.

Following the comments above, I wonder if the authors could also discuss that they are mainly working on the classification scenario where a proper scoring rule is defined, particularly for the Bayesian setting.

**Strengths And Weaknesses:**

(Originality): While modelling aleatoric and epistemic uncertainties has gained much interest in recent years, to the reviewer's knowledge, this paper is the first work demonstrating the problems on objective functions of related modelling approaches. In particular, the authors gave a systematic view of the problems that arise from both Bayesian and non-Bayesian treatments.

(Significance): This paper provides a sounding argument for major approaches that claim to offer reasonable epistemic estimates. Therefore the reviewer expects this paper to change the view of the uncertainty quantification community. The message is relatively straightforward: (1) the averaged (proper) loss minimisation shouldn't be used for epistemic uncertainty. (2) the prior / regularisation setting might also lead to inappropriate epistemic uncertainty for a Bayesian setting.

(Quality, Clarity): This paper comes with systematic mathematic notations and formalisations. Readers with similar research interests should have no issues following the discussion. The technical statement is motivated by clear definitions as well as working examples.

On the weaknesses:

(Quality) Given that the paper is pointing out the problem of existing methods, it is understandable that the paper doesn't come with a pack of numerical experiments. However, it would still benefit the readers with a few demonstrated experiments. Even if there is no ground truth for the level-2 uncertainty, both requirements in definition-1 should be able to be examined with some synthetic data distributions. Therefore some demonstrative experiments with different data sizes and loss functions should help the community to receive the paper better.

---

> ### Author Response · Authors · 2022-08-02
> **Response to Reviewer MUhd**
>
> First of all, thank you very much for appreciating the quality, and originality of our paper as well as the significance of our results. In the following, we would like to address some of the concerns as well as the question raised by the reviewer.
>
>
> ### Weakness: Even if there is no ground truth for the level-2 uncertainty, both requirements in definition-1 should be able to be examined with some synthetic data distributions.
>
> Due to page restrictions, we have given only one experiment for illustration. However, as the accepted papers are allowed an additional content page, it would be no problem to extend the experiments. Thanks for the suggestion.
>
> ### Questions: While the discussion on the (Average Level-1 Loss) is clear: the reviewer found the debate on (Bayesian Losses and Regulations) requires an additional assumption that the uniform posterior gives the maximal uncertainty. I wonder if the authors can comment if such an assumption is too restrictive?
> ### (1) In a regression setting where the posterior has infinite support, it might be hard to come up with such a uniform distribution without additional transformation.
> ### (2) Also, for a highly unbalanced classification problem, the uniform distribution might not be the case for maximal uncertainty.
>
>
> The uniform distribution is commonly considered to reflect maximal uncertainty, both in probability and information theory, and this is quite well justified (e.g., by Laplace’s principle of insufficient reason, axiomatic characterizations of information measures such as entropy, etc.).
>
> That said, it’s true that a uniform distribution cannot be specified in the case of infinite support. This is a general problem in Bayesian inference. One way out is to work with an improper distribution (e.g., improper prior), which, however, is not always possible. Then, the only alternative is to deviate from the uniform distribution.  For example, in the regression setting considered by Amini et al. [2020] in “Deep Evidential Regression”, the authors assume a noisy Gaussian observation model, which then quite naturally leads to a Normal Inverse-Gamma distribution over the parameters of the Gaussian.
>
> As for (2), one has to distinguish between aleatoric (level-1) and epistemic uncertainty (level-2). On the first (aleatoric) level, where we consider distributions over class labels $y$, please note that these distributions are conditional distributions $p(y|x)$. Therefore, global class imbalance over the entire instance space $\mathcal{X}$ is not of direct relevance here. Instead, the distribution with the highest aleatoric uncertainty is again the uniform distribution, as the learner is as undecided about the class label to predict as it could be. On the second (epistemic) level, the distribution is a distribution over (class) distributions representing model uncertainty and reflecting the epistemic state of the learner. Here, it is essentially the same: A uniform level-2 distribution gives each level-1 distribution the same probability (or belief), thereby representing maximal uncertainty.
>
> ### Limitations: Following the comments above, I wonder if the authors could also discuss that they are mainly working on the classification scenario where a proper scoring rule is defined, particularly for the Bayesian setting.
>
> Thank you for pointing this out! We will make this clearer.

---

> > ### Comment · Reviewer_MUhd · 2022-08-08
> > **Follow-up comments**
> >
> > Thank the authors for the feedback.
> >
> > (1) Over the synthetic experiments, I wonder if the authors can provide some initial design for the experiments. Given such experiments should be pretty helpful in conveying the ideas from this paper, it would be good to hear more before the final publication (if accepted).
> >
> > (2) I am still not sure if I totally agree with "global class imbalance over the entire instance space is not of direct relevance here". Consider the problem of predicting a very rare disease. Given a new doctor/learner that is very uncertain about an unfamiliar case, I think it also makes sense to make a prediction according to the prior disease probability for level-1 (i.e. not sure about this case but unlikely to have the disease according to the general population), while the level-2 could be a distribution of binary probabilities with a high variance but shares a marginal distribution close to the prior disease probability.  I guess this is more about how we expect the model to behave when seeing a data point that is far away from the training points. In the binary case, I think both (predicting 0.5) and (predicting class prior) make sense.

---

> > > ### Author Response · Authors · 2022-08-09
> > > **RE: Follow-up comments**
> > >
> > > Ad (1). We would consider four representative scenarios: two in a binary classification and two in a multiclass setting.
> > > In one of the two scenarios the aleatoric uncertainty is high (or maximal), i.e., a uniform distribution as the ground-truth level-1 distribution, while in the other the aleatoric uncertainty is rather low, i.e., a concentrated level-1 distribution (this covers the imbalanced learning scenario).
> > > We would generate repeatedly observations of different sizes $N$ and compute the corresponding ERM for the two different classes of level-2 loss functions based on cross-entropy or Brier score in each run.
> > > We would report/plot the mean entropy (together with the standard deviations) of the ERM’s in dependence of the data set size $N$ for different values of $\lambda$.
> > >
> > > For the binary classification scenario with the highest aleatoric uncertainty ($p(y)\sim Ber(0.5)$) we have the following results for using the Brier score as the level-1 loss
> > >
> > > |     | N=10 | N=50 | N=100 | N=200 | N=500 |
> > > |------------------|------|------|-------|-------|-------|
> > > | $\lambda = 0$    | 0  (0)   |  0  (0)    |   0  (0)    |    0  (0)   |   0  (0)    |
> > > | $\lambda = 0.1$  | 0.158  (0.0831)     | 0  (0)   0  (0)  | 0  (0)      |   0  (0)    |   0  (0)    |
> > > | $\lambda = 0.25$ | 5.0894 (0.0343)     |   0  (0)   |  0  (0)     |  0  (0)     |    0  (0)   |
> > > | $\lambda = 0.5$  | 5.0859 (0.0421)     |  0.1271 (0.0559)    |    0  (0)    |   0  (0)     |    0  (0)    |
> > > | $\lambda = 0.75$ | 5.2787 (0.2321)     |  1.1731 (0.4014)    |     0  (0)   |   0  (0)     |    0  (0)    |
> > > | $\lambda = 0.9$  | 5.3660 (0.2660 )   |   4.4114 (1.3809)   |   0.0392 (0.0631)    |     0  (0)   |    0  (0)    |
> > > | $\lambda = 1$    | 5.4251 (0.2435)     |   4.0386 (1.6524 )   |  0.1741 (0.0806)     |      0  (0)  |   0  (0)     |
> > >
> > > Note that the level-2 uniform distribution has an entropy of 5.6439 in this case, so we see that the ERM is essentially either a point-mass (entropy of zero) or nearly a uniform distribution (having a large entropy).
> > > Looking at the behavior of the ERM as a function of the amount of data $N$ for a fixed $\lambda,$ we find that the ERM changes quite abruptly from one extreme (maximum uncertainty) to the other (maximum certainty), and there is almost no gradual decrease as one would expect.
> > > This demonstrates that the influence of $\lambda$ is in a sense quite arbitrary, since $\lambda$ in principle simply switches epistemic uncertainty back and forth between the two extremes.
> > >
> > > For the binary classification scenario with a low aleatoric uncertainty ($p(y)\sim Ber(0.05)$) we have the following results:
> > >
> > > |     | N=10 | N=50 | N=100 | N=200 | N=500 |
> > > |------------------|------|------|-------|-------|-------|
> > > | $\lambda = 0$    | 0  (0)   |  0  (0)    |   0  (0)    |    0  (0)   |   0  (0)    |
> > > | $\lambda = 0.1$  | 0  (0)      | 0  (0)   0  (0)  | 0  (0)      |   0  (0)    |   0  (0)    |
> > > | $\lambda = 0.25$ | 0.1341 (0.0190)     |   0  (0)   |  0  (0)     |  0  (0)     |    0  (0)   |
> > > | $\lambda = 0.5$  | 1.8576 (0.2103)     |  0  (0)     |    0  (0)    |   0  (0)     |    0  (0)    |
> > > | $\lambda = 0.75$ | 4.9215 (0.0053)     |  0  (0)     |     0  (0)   |   0  (0)     |    0  (0)    |
> > > | $\lambda = 0.9$  | 4.9384 (0.0071)   |   0  (0)    |    0  (0)    |     0  (0)   |    0  (0)    |
> > > | $\lambda = 1$    | 4.9553 (0.0082)     |   0  (0)    |   0  (0)     |      0  (0)  |   0  (0)     |
> > >
> > > The results for the cross-entropy as the underlying level-1 loss function are very similar, so we do not report them here.
> > > Please note that computing the ERM over all level-2 distributions is computationally costly, so that we couldn’t manage to finish the experiments for the multiclass setting.
> > > However, we are quite certain that the findings will be similar and again in agreement with the main theoretical result of our paper.
> > >
> > > Ad (2). What we meant with “global class imbalance over the entire instance space is not of direct relevance here” is that the “right” level-2 prediction should really depend on the feature information $x$.
> > > Indeed, please note that the (global) level-1 prior, say 95% negative and 5% positive, does not directly translate into per-instance level-2 information.
> > > Instead, the same prior can emerge for very different reasons, e.g.:
> > >
> > > - (i) The (0.95,0.05) distribution may apply to all instances $x$. For example, this is the case when the features x are non-informative, so that $p(y|x)=p(y)$.
> > >
> > > - (ii) The degenerate distribution (1,0) applies to 95% of the instances and the distribution (0,1) to 5% of the instances. This is the case when the features are fully informative and uniquely identify the class.
> > >
> > > In both scenarios, per-instance epistemic uncertainty should be represented by the learner in a quite different way.

---

### Official Review · Reviewer_jJ2C · 2022-07-13

**Rating:** 7
**Confidence:** 2
**Soundness:** 3 good
**Presentation:** 4 excellent
**Contribution:** 3 good

**Summary:**

This paper analysises the loss minimisation in second-order learner for epistemic uncertainty quantification. It shows the proposed loss functions do not incentivise the learner to represent the epistemic uncertainty in a faithful way.

**Questions:**

Are there any potential solutions to address the gap?

**Strengths And Weaknesses:**

The arguments are clear and richful.

---

> ### Author Response · Authors · 2022-08-02
> **Response to Reviewer jJ2C**
>
> First of all, thank you very much for appreciating the quality of our paper as well as the importance of our results. In the following, we would like to address the question raised by the reviewer.
>
>
> ### Question: Are there any potential solutions to address the gap?
>
> Regarding your question: This is difficult to tell, but based on our results so far, we currently believe that there is no level-2 loss function that guarantees a faithful representation of epistemic uncertainty in the setting of direct loss minimisation considered in this paper.
>
> Perhaps a way out is to allow for more general types of losses. For example, one could allow for adjusting the regularization parameter for the Bayesian losses with the number of samples, and then consider the ordering of the level-1 predictions instead. The only problem we see with this approach is that the learner’s uncertainty is again controlled in an external way, i.e., it is subjective due to the choice of the regularisation sequence (cf. our remark at the end of Section 4).
>
> Another direction could be the use of different representations of the epistemic uncertainty, i.e., representations other than a level-2 predictor based on a risk minimisation approach. Indeed, other approaches have been proposed in the literature, but again, it needs to be thoroughly investigated whether the stated epistemic uncertainty is faithful.

---

### Meta-Review · Area_Chair_tSpK · 2022-08-24

**Recommendation:** Accept
**Confidence:** Certain

**Metareview:**

This meta review is based on the reviews, the authors rebuttal and the discussion with the reviewers, and ultimately my own judgement on the paper. There was a consensus that the paper contributes interesting insights on uncertainty quantification, and most reviewers praised several aspects of the submission. I feel this work deserves to be featured at NeurIPS and will attract interest from the community. I would like to personally invite the authors to carefully revise their manuscript to take into account the remarks and suggestions made by reviewers. Congratulations!

**Award:**

No

---

### Decision · Program_Chairs · 2022-09-14

Accept